# A single-cell pan-cancer analysis to show the variability of tumor-infiltrating myeloid cells in immune checkpoint blockade

Weiyuan Li[1,2,3,16], Lu Pan [4,16], Weifeng Hong[5,6,7,16], Florent Ginhoux [8,9,10], Xuan Zhang[11], Chunjie Xiao[1] ✉ & Xuexin Li [12,13,14,15] ✉

Myeloid cells are vital components of the immune system and have pivotal functions in orchestrating immune responses. Understanding their functions within the tumor microenvironment and their interactions with tumor-infiltrating lymphocytes presents formidable challenges across diverse cancer types, particularly with regards to cancer immunotherapies. Here, we explore tumor-infiltrating myeloid cells (TIMs) by conducting a pan-cancer analysis using single-cell transcriptomics across eight distinct cancer types, encompassing a total of 192 tumor samples from 129 patients. By examining gene expression patterns and transcriptional activities of TIMs in different cancer types, we discern notable alterations in abundance of TIMs and kinetic behaviors prior to and following immunotherapy. We also identify specific cell-cell interaction targets in immunotherapy; unique and shared regulatory profiles critical for treatment response; and TIMs associated with survival outcomes. Overall, our study illuminates the heterogeneity of TIMs and improves our understanding of tissue-specific and cancer-specific myeloid subsets within the context of tumor immunotherapies.

The development of immune-checkpoint blockade (ICB) immunotherapy marked a significant breakthrough in the field of cancer treatment, yielding promising clinical outcomes in recent years[1,2]. While T lymphocytes have been the primary focus of ICB research, myeloid cells, have not received comparable attention despite their abundance and versatility within the tumor microenvironment

(TME)[3–5]. There is growing interest in investigating tumor-infiltrating myeloid cells (TIMs), but their influence within the TME and their impact on ICB responses remain incompletely understood[5–9]. Currently, Myeloid cells are broadly classified into subgroups including macrophages, monocytes, classic dendritic cells (cDCs) and plasmacytoid DCs (pDCs)[4]. However, this conventional classification often

[1]School of Medicine, Yunnan University, Kunming, Yunnan 650091, China. [2]Department of Reproductive Medicine, The First People's Hospital of Yunnan Province, Kunming, Yunnan 650032, China. [3]The Affiliated Hospital of Kunming University of Science and Technology, Kunming, Yunnan 650031, China. [4]Institute of Environmental Medicine, Karolinska Institutet Solna 171 65, Sweden. [5]Department of Radiation Oncology, Zhejiang Cancer Hospital, Hangzhou 310005, China. [6]Hangzhou Institute of Medicine (HIM), Chinese Academy of Sciences Hangzhou 310005, China. [7]Zhejiang Key Laboratory of Radiation Oncology, Hangzhou 310005, China. [8]Singapore Immunology Network (SIgN), Agency for Science, Technology and Research (A*STAR), Singapore 138648, Singapore. [9]Institut Gustave Roussy, INSERM U1015, Bâtiment de Médecine Moléculaire 114 rue Edouard Vaillant, 94800 Villejuif, France. [10]Shanghai Institute of Immunology, Department of Immunology and Microbiology, Shanghai Jiao Tong University School of Medicine, Shanghai 200025, China. [11]Department of Colorectal Surgery, Yunnan Cancer Hospital, The Third Affiliated Hospital of Kunming Medical University, Kunming, Yunnan 650032, China. [12]Department of General Surgery, The Fourth Affiliated Hospital, China Medical University Shenyang 110032, China. [13]Key Laboratory of Precision Diagnosis and Treatment of Gastrointestinal Tumors, Ministry of Education, China Medical University, Shenyang 110122 Liaoning, China. [14]Institute of Health Sciences, China Medical University, Shenyang 110122 Liaoning, China. [15]Department of Physiology and Pharmacology, Karolinska Institute Solna 171 65, Sweden. [16]These authors contributed equally: Weiyuan Li, Lu Pan, Weifeng Hong. ✉e-mail: chjxiao@ynu.edu.cn; xuexin.li@ki.se

oversimplifies the heterogeneity of TIMs. Numerous myeloid sub-categories have been identified in the TME[3,4], for instance, these include tumor-associated macrophages (TAMs) that are traditionally divided into M1 macrophages, which generally suppress tumor growth, and M2 macrophages, which tend to promote tumorigenesis[3,10,11]. Broad classification fails to capture the full complexity of TIMs. Therefore, instead of identifying myeloid cells as a simple classification, contemporary single-cell research perceives their functional states as a pattern of heterogeneities[3–5,12–14].

Recent advancements in single-cell RNA sequencing (scRNA-seq) have greatly enhanced our ability to observe and decipher the heterogeneities and intercommunications among different subtypes of TIMs within the TME[5,6,8,9]. In this study, we conducted a comprehensive pan-cancer scRNA-seq analysis of myeloid cells using 192 samples from eight distinct cancer types, aiming to elucidate the spectrum of TIM phenotypes and functionalities in both treatment-naïve and post-treatment patients. Importantly, our access to post-treatment clinical response data for five of these cancer types, enabled us to compare pre- vs. post-treatment and therapy-responsive vs. non-responsive groups at a pan-cancer level. We identify distinct subtypes of TIMs and document dynamic shifts in their frequencies, cellular interactions, and regulatory elements across cancer types; these changes are evident between pre- and post-treatment groups and between responders and non-responders to ICB therapy. Integrating our data with clinical insights, we identify critical TIM-to-TIM and TIM-to-T cell targets associated with immunotherapy response status. Furthermore, we discover unique and shared regulatory profiles that play essential roles in determining treatment response, establishing significant correlations between specific TIMs and survival outcomes. By utilizing a specially constructed response index, we achieve accurate separation

of immunotherapy response status. We validate our findings through immunofluorescence analysis of colorectal cancer samples with a particular focus on examining FOLR2+APOE+ macrophages, which have previously demonstrated a positive prognostic association in breast cancer[15]. In this work, our extensive pan-cancer analysis serves as a valuable resource that advances our understanding of the clinical implications of TIM alterations in ICB. The knowledge derived from these findings holds tremendous potential for clinical applications, including the development of more effective therapeutic strategies and personalized treatment approaches in cancer immunotherapy

## Results

### TIM landscape varies significantly among cancer types and treatment groups

To begin assessing TIM heterogeneity across cancers, we conducted scRNA-seq on tumor samples from 129 patients across eight distinct cancer types: basal cell carcinoma (BCC), breast cancer, clear cell renal cell carcinoma (ccRCC), colorectal cancer (CRC), hepatocellular carcinoma (HCC), head and neck squamous cell carcinoma (HNSCC), intrahepatic cholangiocarcinoma (iCCA), and melanoma. Of these, five cancer types (BCC, ccRCC, CRC, HNSCC, and melanoma) had available clinical information regarding treatment response status (Fig. 1a and Supplementary Data 1). Quality control measures were implemented to remove potential debris, damaged cells, and doublets. Batch effect was corrected and assessed before and after the correction process using the LISI index[16] (Supplementary Fig. 1a-b). A total of 47,750 myeloid cells were included for subsequent analysis (Supplementary Fig. 1c). Precise categorization of cells was achieved through manual annotation utilizing well-defined signature markers[6,14,17]. Within the eight cancer types, we identified 12 subtypes of macrophages and

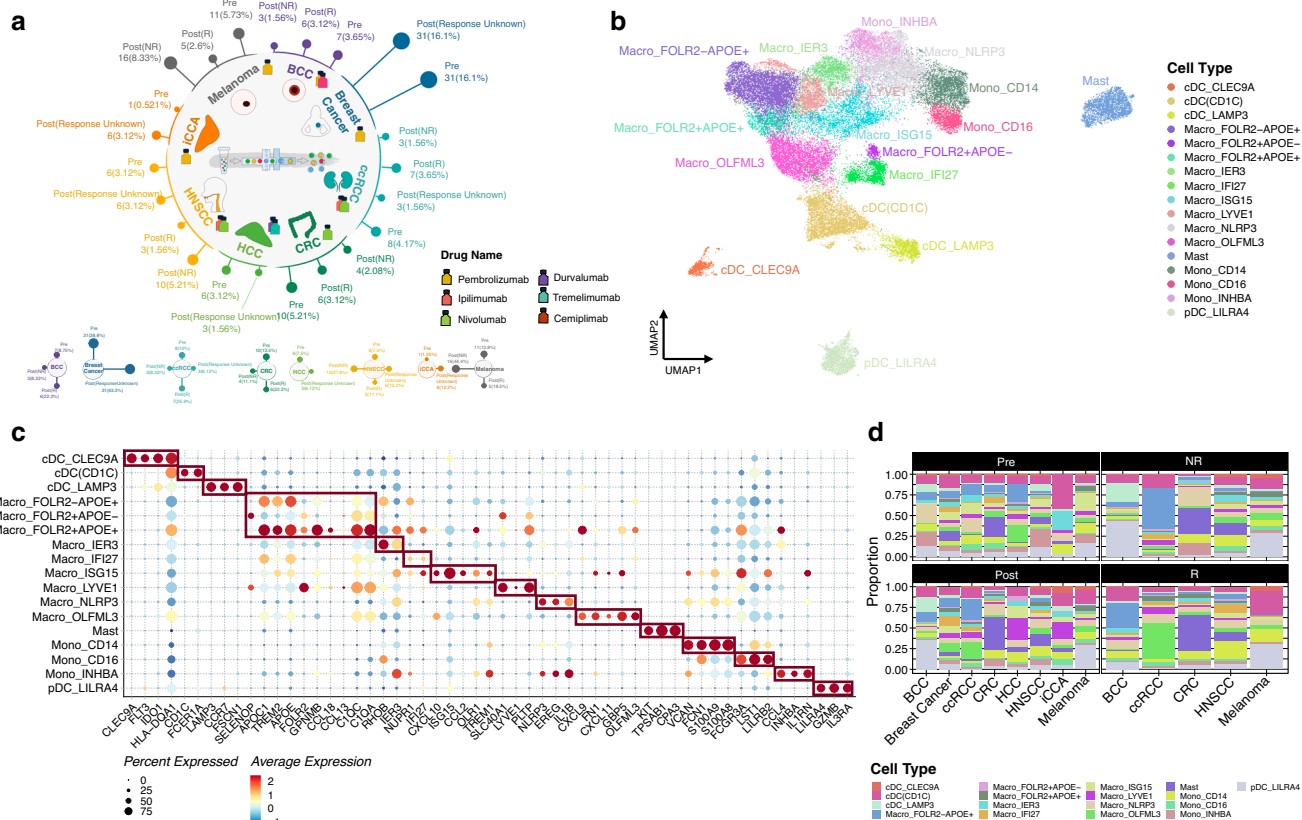

**Fig. 1 | Demographics overview of the pan-cancer myeloid atlas. a** Sample numbers of each cancer type collected; numbers shown in the brackets indicated the sample number in terms of its percentage share in the whole atlas. **b** UMAP

representation of the myeloid atlas. **c** Expression of the signature markers in each cell type. **d** The proportion of cell types across cancer types in treatment-naïve and post-treatment groups.

monocytes, four subtypes of cDCs, and a distinct group of mast cells (Fig. 1b-c and Supplementary Data 2). We differentiated three macrophage subtypes mainly by their expression of FOLR2 and APOE. Other markers were also enhanced, including APOC1 and TREM2 in Macro-FOLR2-APOE+ and Macro_FOLR2 + APOE +; SELENOP in Macro-FOLR2 + APOE-; and GPNMB, CCL18, CCL13, C1QC, and C1QA in Macro_FOLR2 + APOE+ (Fig. 1c). Two clusters of cDCs were clearly separated from the plasmacytoid dendritic cell (pDC) cluster based on exclusive expression of LILRA4, GZMB, and IL3RA (Fig. 1c). HLA-DQA1 was expressed by both cDC clusters, with higher average expression in the cDC_CLEC9A cluster as previously observed[6]. Mast cell-specific markers were also exclusively expressed by this cell population (Fig. 1c).

We further characterized the functionality of these 17 identified cell types (Supplementary Fig. 2a). Macro_APOC1 + IFI27 + , Macro_ISG15, and Macro_OLFML3 exhibited functional differences despite their high expression of CXCL9 and CXCL11. Macro_APOC1 + IFI27+ primarily engaged in GTPase-activating protein binding, Macro_ISG15 was predominantly enriched in cytokine receptor binding and Macro_OLFML3 may participate in fatty acid synthase activity. Additionally, within the three macrophage types defined by FOLR2 and APOE, Macro_FOLR2 − APOE+ was mainly associated with "protein-lipid complex binding," while Macro_FOLR2 + APOE− and Macro_FOLR2 + APOE+ were respectively linked to "protein serine/threonine/tyrosine kinase activity" and "phosphatidylinositol binding" (Supplementary Fig. 2a).Considering the signature markers for M1/M2 macrophages[11,18], Macro_FOLR2 + APOE+ was mainly enriched for M1 markers and Macro_LYVE1 for M2 markers, while Macro_ISG15, Macro_NLRP3, and Macro_FOLR2 + APOE+ were enriched for both. Conversely, Macro_OLFML3, Macro_APOC1 + IFI27 + , Macro_FOLR2-APOE + , and Macro_FOLR2 + APOE- did not show enrichment for either M1 or M2 markers (Supplementary Fig. 2b).

We also utilized the Role index to analyze the preferential distribution of the 17 cell types between responders and non-responders to ICB therapy, finding that Macro_ISG15 (associated with interferon), Macro_FOLR2 + APOE+ (with high CXCL9 expression), and cDC_CLEC9A (high IDO1 expression) exhibited a preference for the responder group (Supplementary Fig. 2a−c). Surprisingly, we observed a preference for the non-responder group in cDC (CD1C) and cDC3_LAMP3 (Supplementary Fig. 2c). This suggests that although DCs are involved in antigen presentation, an increase in their abundance may represent increased anti-tumor cytotoxicity and immune awakening. Overall, the above analysis highlights the limitations of the M1 and M2 classification in capturing macrophage heterogeneity, emphasizing the need for a more accurate classification system that incorporates the diversity and dynamic functional characteristics of macrophage subsets.

The samples from the eight cancer types were categorized based on their treatment status, including treatment-naïve (Pre), post-treatment (Post), treatment non-responders (NR), and treatment responders (R). Subsequently, we compared the distribution of cell types between each pair of stratified groups to assess any variations (Fig. 1d, e and Supplementary Fig. 3a, b). In the Pre group, cDC (CD1C) was the predominant cell type present in iCCA and breast cancer. Macro_NLRP3 accounted for the largest proportion in BCC and ccRCC. In the Post group, pDC_LILRA4 became the main cell type in both BCC and melanoma, while Macro_LYVE1 was predominant in HCC and iCCA (Fig. 1d). Notably, mast cells remained the major cell type in CRC in both the Pre and Post groups. Mast cells were consistently the major cell type in CRC across the four stratified groups (Fig. 1d). Several cancer types exhibited significant decreases in the proportions of Macro_ISG15 and Mono_INHBA after treatment, while Macro_FOLR2 − APOE + , Macro_APOC1 + IFI27 + , Macro_LYVE1, and Macro_OLFML3 showed significant increases in the Post group (Fig. 1d, e and Supplementary Fig. 3a-b). Comparing the response groups, the proportions

of Macro_NLRP3 and Mono_CD14 significantly increased in the R group across many cancer types, whereas no obvious proportional changes were observed in the NR group (Fig. 1d, e and Supplementary Fig. 3c, d). We also analyzed the expression of immunoinhibitors and immunostimulators across various cell types, finding that Macro-FOLR2-APOE+ exhibited minimal expression of immune checkpoint genes, whereas Macro-FOLR2 + APOE+ displayed high expression levels of several immunoinhibitors and immunostimulators, including CTLA4 and CD274 (PD-L1) (Supplementary Fig. 3e).

This analysis collectively highlights notable variations in the composition of TIMs among different cancer types, emphasizing the heterogeneity of these cells across various cancers, treatment statuses, and response groups.

## Treatment and response statuses are associated with different TIM kinetic profiles

The TME is a complex milieu consisting of diverse cell types that undergo continuous dynamic shifts[19–21]. To investigate temporal changes and developmental patterns within TIMs, we conducted a comprehensive analysis of their kinetic profiles at the pan-cancer level by examining the pseudotime trajectories (a measure of differentiation progression) of each individual cell type. We compared Pre vs. Post and NR vs. R samples. Correlation analysis revealed that the TME underwent significant remodeling following immunotherapy. There were notable changes in the intercellular correlations, including a substantial increase in the correlation between Mono_CD14 and Macro_NLRP3 after immunotherapy (Supplementary Fig. 4a). We next employed Slingshot[22] to determine the topological heterogeneity of the differentiation trajectories of macrophages/monocytes and DCs (Supplementary Fig. 4b). Following immunotherapy, distinct alterations were observed in the average pseudotime of specific cell types. Mono_CD14, Macro_NLRP3, Mono_CD16, and Macro_IER3 exhibited significant shortening of their average pseudotime, which would suggest a less differentiated state in the post-treatment. Conversely, pDC_LILRA4 showed a notable delay in trajectory differentiation following immunotherapy (Fig. 2a, b). Meanwhile, cDC (CD1C), cDC_LAMP3, mast cells, Macro_LYVE1, and Macro_FOLR2 + APOE+ displayed minimal changes in average pseudotime between Pre and Post samples (Fig. 2a, b). Mono_CD14, Macro_IER3, Macro_NLRP3, Macro_ISG15, and cDC_CLEC9A displayed considerable delays in pseudotime in the NR compared to the R group, while no substantial kinetic differences were detected in mast cells, Macro_IFI27, and Macro_OLFML3 in these groups (Fig. 2b, c). Interestingly, we observed more cell types with large kinetic changes in the R vs. NR groups than in the Pre vs. Post groups (Fig. 2a, b), suggesting a potential transitional difference between homeostatic and non-homeostatic differentiation in different disease statuses. Notably, Macro_FOLR2 + APOE+ displayed an expedited pseudotime in the R group, in stark contrast to Macro_FOLR2-APOE+ and Macro_FOLR2-APOE- that displayed the reverse pattern despite expressing only one of the markers (Fig. 2a, b).

Our next aim was to elucidate the prevailing biological states within stratified disease cohorts. To achieve this, we classified cells based on their kinetic states and established correlations between the differentially expressed genes in these clustered kinetic profiles and the hallmark gene sets from the Molecular Signatures Database (MSigDB)[23]. We observed distinct branching of pro- or anti-inflammatory signals (Fig. 2c, d and Supplementary Data 3). Within the macrophage and monocyte (Macro/Mono) populations of the R group, the majority of pseudotime states exhibited down-regulation of pro-inflammatory pathways (Fig. 2c), such as IFN-α/γresponse, TNF signaling via NF-κB, and oxidative phosphorylation[24] (States 1 and 2), indicating the presence of an anti-inflammatory signaling milieu. However, State 3 exhibited an upregulation of TNF signaling, suggesting that a minority of cells still promote pro-inflammatory

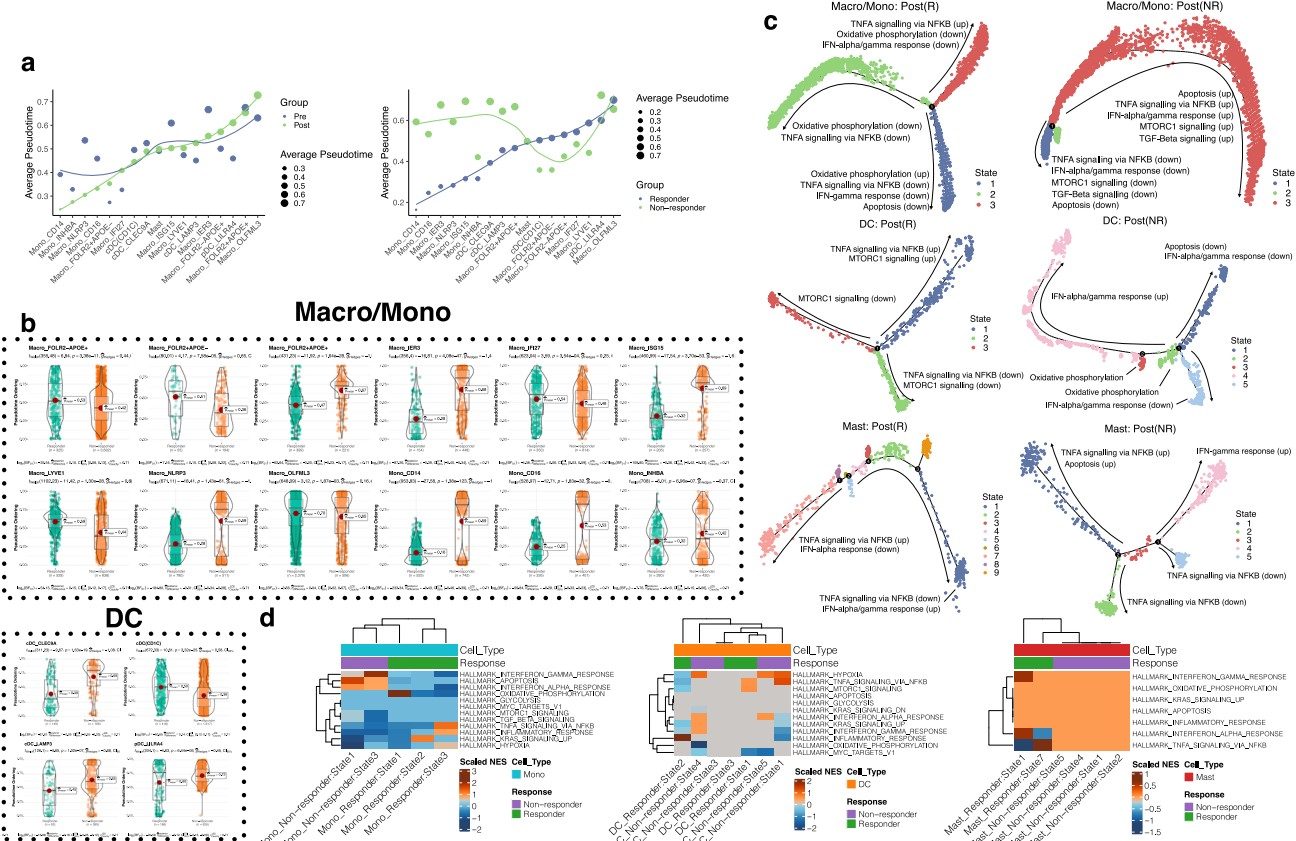

**Fig. 2 | Pseudotime transitions across TIMs. a** Average pseudotime of each cell type in the treatment groupsand the response groups. **b** Violin plots showing the distribution of Pseudotime ordering for responders and non-responders in each cell type. Data are presented as mean values ± SEM (Standard Error of the Mean), indicated by the red dots and associated error bars. The box plots within the violin plots represent the interquartile range (IQR), with the center line indicating the median. The whiskers extend to the minimum and maximum values within 1.5 times the IQR from the 25th and 75th percentiles. Statistical significance was determined using two-sided *t*-test. Multiple comparison adjustment was carried out using BH FDR. *P*-values for each comparison between responder and non-responder groups in each cell type are labeled in the plots, shown under the names of the cell types as "*P* = (*p*-value)". **c** Enriched pathways in each of the pseudotime branches for the three categories of cells, namely Macro/Mono (representing macrophages and monocytes), DC, and Mast. **d** Normalized enrichment scores (NES) of the enriched hallmark pathways in each pseudotime branch.

signaling (Fig. 2c, d, Supplementary Fig. 4b, and Supplementary Data 3). Conversely, in the NR group of Macro/Mono populations, the majority of the cells existed in a pro-inflammatory state (State 3), including cells demonstrating elevated mTORC1 signaling[25].

Similarly, within the DC subpopulations, two states in the R group (States 2 and 3) exhibited anti-inflammatory properties, while State 1 displayed a pro-inflammatory milieu (Fig. 2c, d). In State 1, the majority of cells were identified as cDC (CD1C), with a small subset represented by pDC_LILRA4 (Supplementary Fig. 4 and Supplementary Data 3). Conversely, in NR, pro-inflammatory oxidative phosphorylation was observed in States 2 and 3. State 4 demonstrated upregulation of the IFN-α/γ response (Fig. 2c, d), which was downregulated in States 1 and 5 (Fig. 2c, d). The pro-inflammatory State 4 primarily consisted of cDC (CD1C), alongside a small subset of pDC_LILRA4, mirroring the observations made in the R group (Supplementary Fig. 4b). Meanwhile, the kinetic states of mast cells revealed distinct clusters in the R and NR groups (Fig. 2c, d, Supplementary Fig. 4b and Supplementary Data 3). In the R group, co-upregulation of both anti- and pro-inflammatory signaling was observed within the state clusters. However, in the NR group, a clear separation between pro-inflammatory (States 1 and 4) and anti-inflammatory state clusters (States 2 and 5) was discerned (Fig. 2c, d). These varying kinetic profiles indicate the plasticity of TIMs in the TME and suggest their potential roles as either pro- or anti-pathogenic cell types in the context of immunotherapy.

## Distinct TIM regulatory states characterize specific cancers
We subsequently analyzed gene-regulatory networks to identify active regulatory elements (REs) within TIMs from the R and NR groups. By assessing the transcriptional activity of transcription factors (TFs) in each cell type, we measured correlations between TIMs across different cancer types and response statuses. As shown in Fig. 3a, cell types originating from the same cancer and exhibiting the same response status clustered together. We also observed a close relationship between each R and NR group with their respective cancer types, indicating the high cancer-specificity of REs in TIMs in relation to immunotherapy response. We then identified TF modules contributing to this clustering phenomenon (Fig. 3b and Supplementary Data 4). Notably, elevated YBX1 activity was observed in Macro_FOLR2 + APOE +, Macro_FOLR2 + APOE-, Macro_APOC1 + IFI27 +, cDC(CD1C), Macro_OLFML3, and cDC_CLEC9A in responders with CRC, while BCL3 activity was increased in Mono_CD14, Macro_NLRP3, Mono_INHBA, Macro_OLFML3, and Macro_ISG15 in non-responders with CRC. Intriguingly, the identified TFs were predominantly associated with macrophages. For instance, YBX1 depletion in mouse macrophages has been associated with augmented tissue damage, myofibroblast activation, and fibrosis. MAFB activity was consistently decreased in the R vs. the NR groups across various tumor-infiltrating immune cell categories, including Macro/Mono, DC, and Mast categories (Fig. 3c). MAFB was previously found to promote inflammation in classical TIMs and EMT in lipid-associated TIMs[26,27]. Finally, within the Macro/Mono

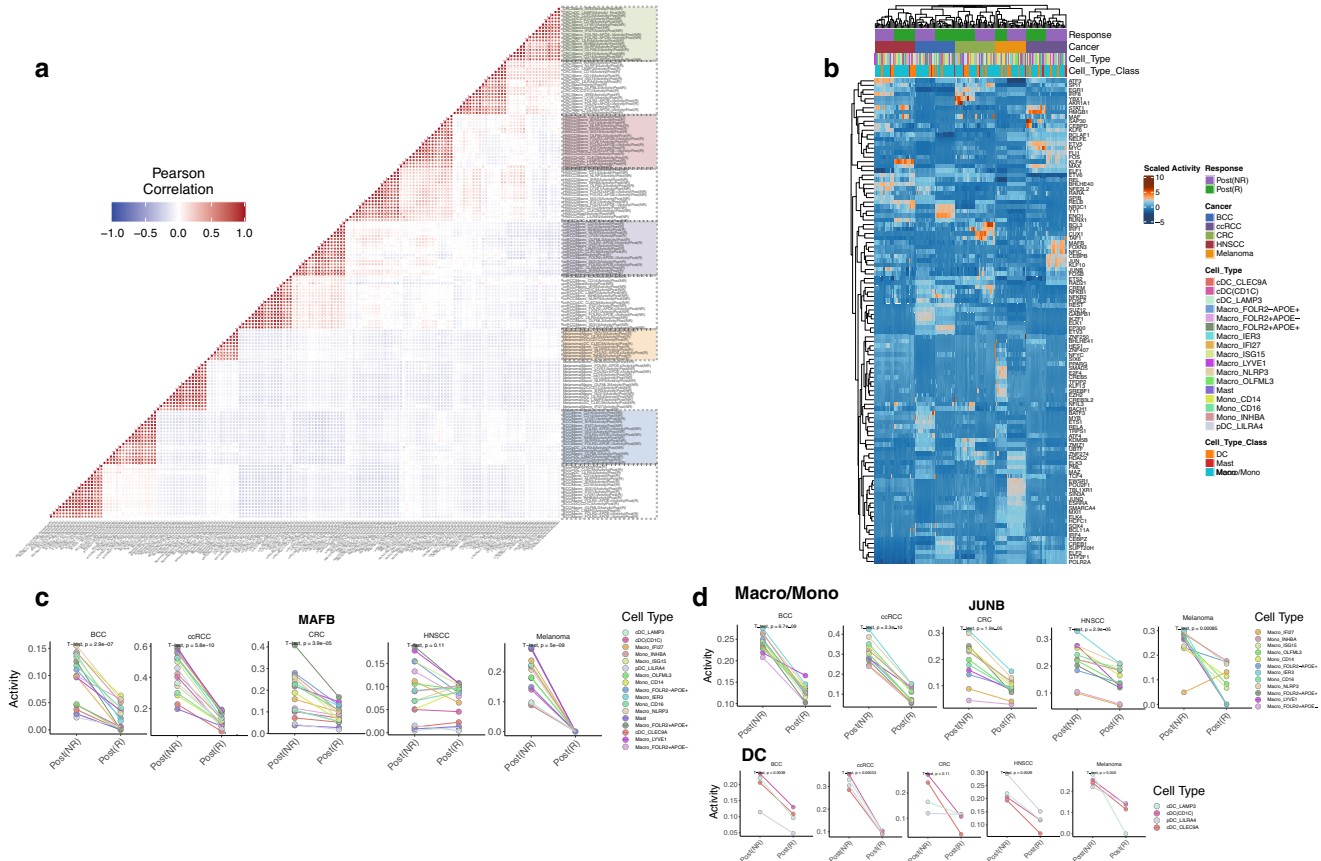

**Fig. 3 | Cell-type-specific regulon activity landscape. a** Correlation of the cell types across cancer types in the response groups based on TF activity. **b** TF modules forming the correlation patterns in (**a**). **c** TF with common activity patterns across cancer types and across cell types in the response groups. For each cancer type and regulon, we performed two-tailed paired *t*-test to compare the activity scores of Post(R) and Post(NR) of each regulon across 17 different cell types, with no technical replicates involved. **d** TFs with common activity and expression patterns across cancer types in the response groups of Macro/Mono and DC categories.

and DC categories, the pro-inflammatory TF JUNB[19] was less active in the R compared to the NR group (Fig. 3d).

## Prospective targets of cell-cell interaction in ICB response

The intricate dynamics of cell-cell interaction within the TME involve not only TIMs but also other immune cell populations, particularly T cells. We examined both intercommunication between TIMs (Fig. 4a, b, Supplementary Figs. 5, 6, and Supplementary Data 5) and the interactions between TIMs and CD4/CD8 T cells (Fig. 4a, b, Supplementary Figs. 5, 6, and Supplementary Data 5) and the interactions between TIMs and CD4/CD8 T cells (Fig. 4c, d, Supplementary Figs. 7–9, and Supplementary Data 6), utilizing the same set of collected samples. Specifically, we analyzed 272,017 T cells from the post-treatment samples and annotated 12 subtypes of CD4 T cells and 15 subtypes of CD8 T cells based on their distinctive molecular signatures (Supplementary Fig. 7).

We applied rigorous criteria for selecting ligand-receptor interactions, considering those with an aggregate rank <0.01 and cell-specific interaction frequencies exceeding the 75th percentile of overall frequencies for each sample condition (see Methods). As shown in Fig. 4a, b and Supplementary Fig. 5, noticeable differences emerged in the interactions between TIMs within the R and NR groups. Notably, Mono_CD16, Macro_IER3, Macro_FOLR2 + APOE + , Macro-FOLR2 + APOE-, and cDC_LAMP3 exhibited substantially higher interactions in the NR group across all cell types, whereas Macro_FOLR2-APOE+ displayed elevated fold ligand interactions with other TIMs in the R group. Additionally, we frequently observed HLA-A/B/C to LILRA1/3 interaction pairs specific to the NR group, demonstrating

increased interaction fold changes across various cell types (Supplementary Fig. 6). As previously reported[28–31], this interaction may indicate a myeloid deactivation and reduced immune response.

Subsequently, we investigated interactions between TIMs and specific CD4/CD8 T cell subsets. Compared to TIMs-to-TIMs interactions, fewer myeloid cell types exhibited large fold changes (absolute fold change > 2) in TIMs-to-CD4/CD8 T cell interactions (Fig. 4c, d, Supplementary Data 6, and Supplementary Fig. 7). Nonetheless, we observed high interactions of pDC_LILRA4 with CD4 T cells; of Mono_CD16, Macro_IFI27, Macro_FOLR2 + APOE-, and Macro_FOLR2-APOE+ with CD8 T cells; and of Mono_INHBA, Macro_FOLR2 + APOE + , and cDC_LAMP3 with CD4/CD8 T cells (Fig. 4d and Supplementary Fig. 8). The HLA to LAG3 interaction cluster was exclusively present in the ligand interactions of pDC_LILRA4 with CD4(IFNG+ Tfh/Th1) and of Mono_CD16 with mainly CD8(GZMK+ Tex), CD8(GZMK+ Tem), and CD8(ISG + T) in the NR group (Supplementary Fig. 9). Regarding Macro_FOLR2 + APOE+ receptor interactions with CD4 T cells, we observed the unique presence of RPS19 to C5AR1 interaction in the NR group, which has previously been shown to suppress antitumor immune response through the production of immunosuppressive cytokines like TGF-β[32]. CD99 to PILRA/CD81 interactions were also substantially increased in the NR group, CD99 and CD81 have been associated with increased tumor migration, invasion, and metastasis[33,34], while PILRA delivers inhibitory signals related to natural killer (NK) cell and DC activation[35] (Supplementary Fig. 9). Although interactions between TIMs and CD4 T cells are less studied, their positive involvement in pro-tumoral functions can be inferred. We identified unique interactions among members of the TNFRSF

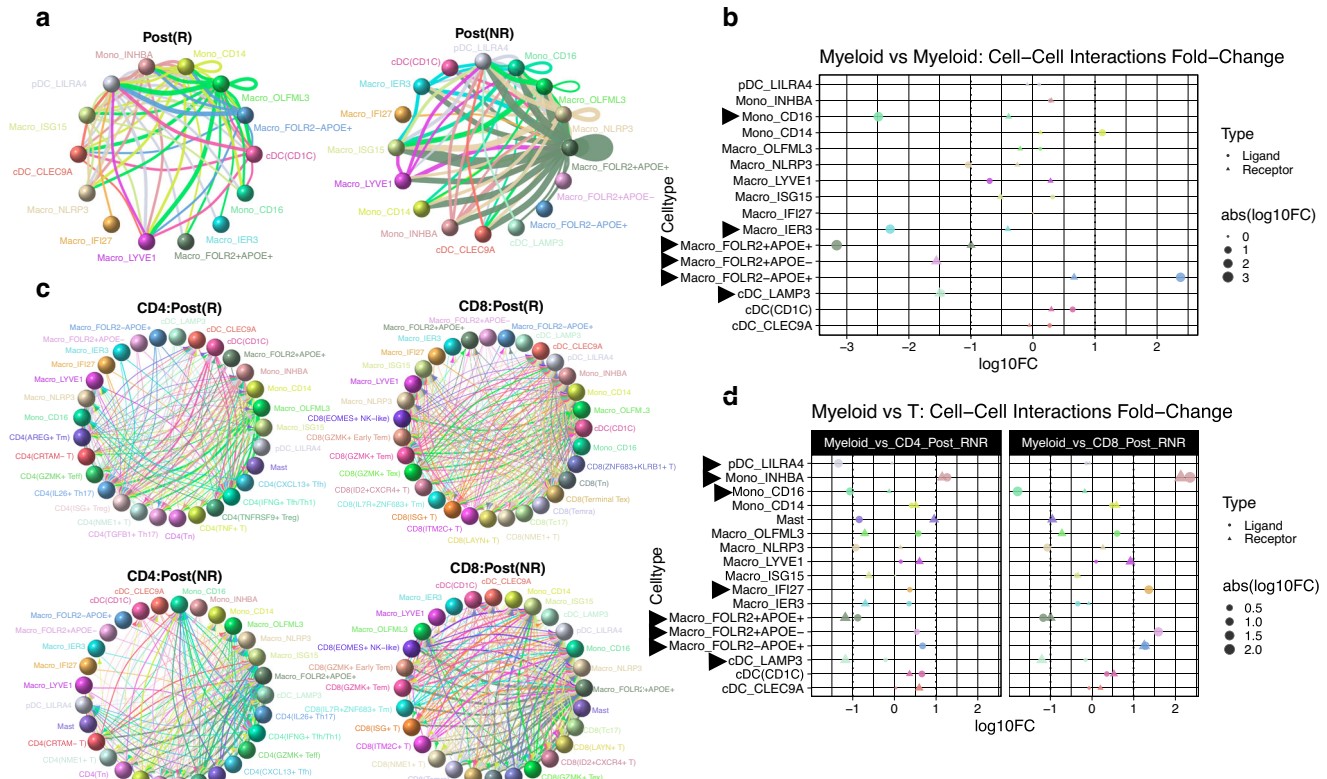

**Fig. 4 | Cell-cell interaction patterns across cell types in response groups. a** Cell-cell interactions between TIMs. Line thickness represents the degree of interactions. **b** Fold change of interaction frequencies between the R (numerator) and NR (denominator) groups. **c** Cell-cell interactions between TIMs and CD4 or CD8 T-cells. Line thickness represents the degree of interactions. **d** Fold change of TIMs and CD4 or CD8 T cells interaction frequencies, between the R (numerator) and NR (denominator) groups.

(i.e., TNFSF10 with TNFRSF11B and LTB with CD40) in the receptor interactions of cDC_LAMP3 with CD4/CD8 T cells in the NR group (Supplementary Fig. 9).

In our analysis of cell types exhibiting higher fold change in the R group, we observed significant ligand interactions involving Mono_INHBA, Macro_APOC1 + IFI27 + , and Macro_FOLR2 − APOE+ with specific subsets of T cells. Notably, HLA to LAG3 interactions were present, primarily interacting with CD4(IFNG+ Tfh/Th1), CD8(GZMK+ Tem), and CD8(Terminal Tex) cells. Additionally, RPS19 to C5AR1 interactions were found in the R group for Mono_INHBA receptor interactions with T cells (Supplementary Fig. 9). The significance of these interaction pairs in relation to response status remains unclear. However, they hold promise as potential targets for further investigation of the plasticity of TIMs and the reciprocal involvement of interactions in both response groups across myeloid-to-T cell interactions.

### Highly interactive TIMs exhibit cancer-specific regulatory patterns

To comprehensively understand the transcriptional regulatory profile of myeloid cells in the context of immunotherapy, we examined the regulatory patterns in cell types that exhibited differential cell-cell interactions between the R and NR groups. Our focus was on the TF signatures of highly interactive TIMs.

We observed differential expression of distinct regulon modules in specific cell types. For instance, regulon module cluster c9 was uniquely expressed by pDC_LILRA4, while c11 and c3 were expressed by cDC_LAMP3. Additionally, several distinct TFs specifically regulated the associated TIM response class (Fig. 5a). In the case of pDC_LILRA4, regulons from cluster c3 and from c6 (excluding REST) were uniquely expressed by the NR group. Comparatively, ELF2, RARA, and TAF1

from c12 displayed distinctive expression patterns in the R group. Other unique signatures differentiating the R and NR groups were predominantly found in the R group, including cluster c4 for Macro_-APOC1 + IFI27+ and c13 for Mono_INHBA. For other cell types, the signature differences between R and NR regulatory expressions were marked by one or more TFs that were not clustered in a regulon module. These regulatory signatures may represent key regulators responsible for the variation in cellular interactions between the R and NR groups of the same TIM type. Aside from the regulon modules, we further identified common TFs across cancer types that regulated in the same direction when comparing the R and NR groups (Fig. 5b). Notable examples include FOSB in Macro_FOLR2 + APOE+, MAFF and SPI1 in cDC_LAMP3, and MAFB in Macro_FOLR2-APOE+.

### Different TIMs are associated with survival in different cancers

To evaluate the impact of individual stratified cell types on patient prognosis, we conducted survival analysis using post-immunotherapy cohorts including the IMvigor210 urothelial carcinoma (UC)[36], ccRCC[37], and skin cutaneous melanoma (SKCM)[38]. Patients were categorized based on the expression of signature markers for the highly interactive TIMs. Our stratification revealed that high expression of cDC_LAMP3 markers correlated with lower survival, whereas high expression of Macro_FOLR2 + APOE-, pDC_LILRA4, and Macro_-APOC1 + IFI27+ markers was associated with improved survival outcomes (Fig. 5c). These markers hold potential for predicting survival outcomes following immunotherapy. However, certain cell types, including Mono_INHBA, displayed opposite survival relationships in different cancer types emphasizing the cancer-specificity and macrophage heterogeneity observed in relation to survival outcomes (Fig. 5c).

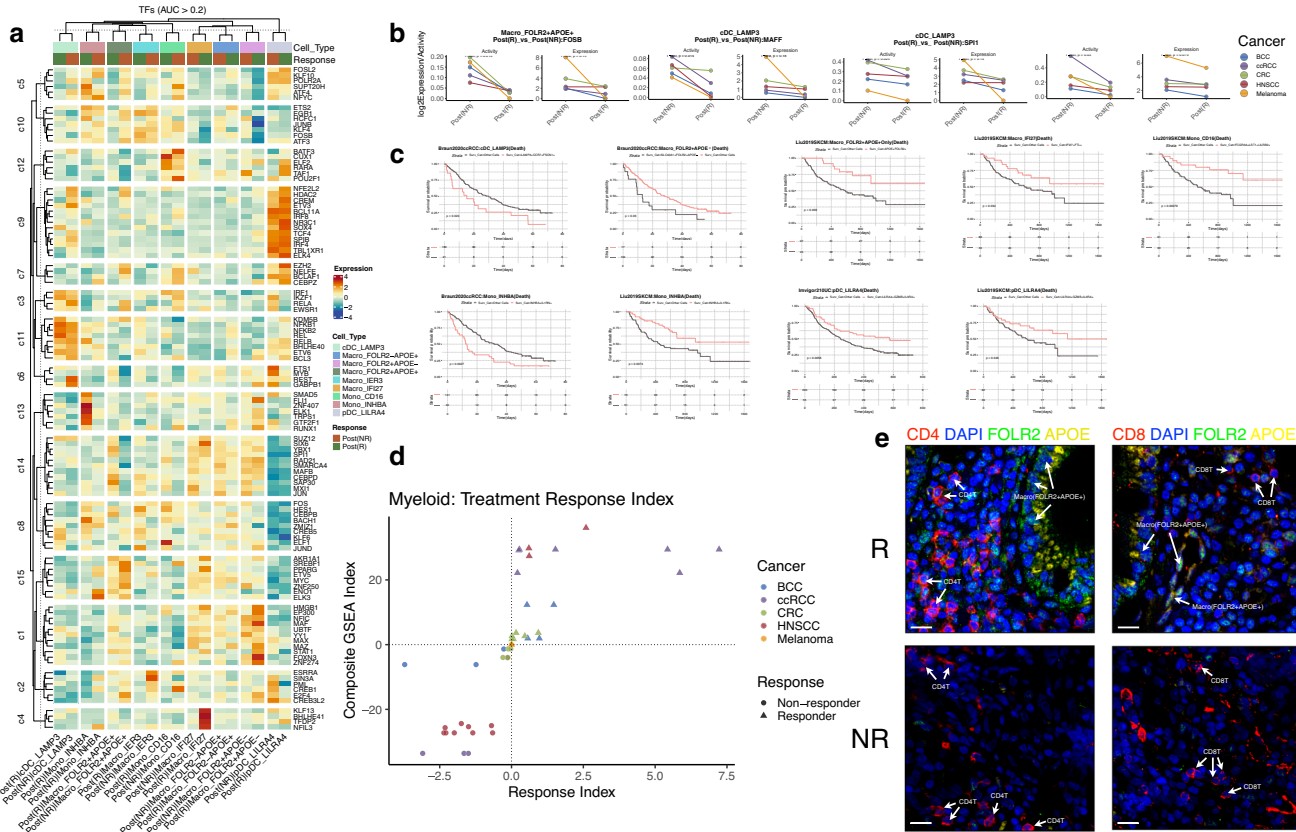

**Fig. 5 | Targeted analysis of the highly interacting TIMs. a** Standardized expression values of the TF signatures of the nine TIMs in the response groups are shown in the plot. Color gradient indicates the standardized expression values of the TFs. **b** TFs with common activity patterns across cancer types in the response groups of these cell types. For each cell type and regulon, we performed two-tailed paired *t*-test to compare the activity scores or the expression values of the regulon between Post(R) and Post(NR) across cancer types. The analysis included five different cancer types, with no technical replicates involved. **c** Survival analysis based on the expression of the signature genes of these cell types. Log-rank test was used to compare the survival distributions within each survival plot. No multiple testing correction was carried out. **d** Response index and GSEA index separating the samples of the two response groups. Each point represents a sample in the study with response information. **e** Multiplex immunofluorescence staining has delineated the spatial relationships between the Macro_FOLR2 + APOE+ cell subpopulation and T cells, as well as their abundance in ICB responders and nonresponders among CRC patients.

After identifying the prominent interaction of HLA-A/B/C with LILRA1/3, primarily within Mono_CD16 cells of the NR cohort, we aimed to elucidate the predictive capability of these interactions. To achieve this, we evaluated the collective expression of HLA-A/B/C, LILRA1, LILRA3, LILRB2, and the signature markers of Mono_CD16 to quantify the cumulative impact of these factors on patient prognosis. The difference in survival outcomes between the two groups expressing Mono_CD16 signature markers was not significant in the UC and ccRCC cohorts (considering LILRB2 as one of the signature markers for Mono_CD16), however, when considering the additional expression of HLA-A and/or LILRA1, the higher-expression group demonstrated significantly lower survival rates in both cohorts (Supplementary Fig. 10). Conversely, higher expression of these markers was associated with better survival rates in the SKCM cohort Consequently, although HLA-A/B/C and LILRA1/LILRA3/LILRB2 were commonly observed in nonresponder TIMs, their association with post- ICB patient survival appears to be cancer-specific.

## Construction of response index and GSEA index which accurately determined treatment response

We subsequently conducted gene ontology (GO) pathway analysis using the top upregulated differentially expressed genes (DEGs) of each TIM cell type in each R group to unravel the underlying changes associated with treatment response. The results revealed distinct immune-related pathway differences between the R and NR groups. In

the R group, Macro_FOLR2 + APOE+ was involved in the NK-mediated immune response; Mono_CD16 showed negative regulation of tumor-promoting NF-κB activity and upregulation of macrophage tolerance induction, which protects the host from chronic exposure to inflammatory mediators[39]; and Mono_INHBA demonstrated positive regulation of both IL-4 and IL-1αproduction, which together trigger anti-tumoral Th9 cell differentiation[40] (Supplementary Fig. 11). Conversely, in the NR group, Macro_FOLR2-APOE + , Macro_FOLR2 + APOE-, and Macro_APOC1 + IFI27+ were associated with apoptotic cell clearance, and Macro_IER3 was involved in the regulation of anti-tumoral IL-15 and IL-12 production. Identifying and understanding the roles of individual TIMs under different response conditions would facilitate the targeting of deleterious factors to improve conditions within the R group.

We next aimed to develop an index specifically designed to predict the response to ICB therapy. This index is based on the enriched pathways in each cell type under different response conditions and on the proportion of each cell type within these conditions for each cancer type. To construct the index, we conducted extensive gene set enrichment analyses (GSEAs) across MSigDB signature categories, encompassing all cell types across different cancers within each categorized group. This allowed us to identify cancer-specific and response-specific signatures for each cell type. By considering the cancer-specific proportions of cell types and the enrichment scores of the identified signatures, we created a treatment response index that

clearly distinguished between ICB responders and non-responders at index = 0 (Fig. 5d). In certain cancer types, such as HNSCC and CRC, distinct clusters of non-responders and both responders and non-responders, respectively, were observed at negative and positive response indices. However, other cancer types displayed a wide dispersion of response indices, indicating greater heterogeneity in treatment response. Notably separation based on the response index was not possible in melanoma due to the absence of identified melanoma-specific signatures. This highlights the importance of considering cancer-specific characteristics when assessing treatment response and the variable predictive performance of the response index across different cancer types.

## Macro_FOLR2 + APOE+ cells are more abundant in ICB-responsive CRC patients

The density of FOLR2+ macrophages in tumors has been correlated with survival in breast cancer and HCC patients[15,41]; therefore, the significant role we observed for Macro_FOLR2 + APOE+, and its interaction with T cells, is notable. To corroborate the alterations in cell type composition within the realm of immunotherapy, we conducted immunofluorescence analysis utilizing samples obtained from ICB responder and non-responder CRC patients. This revealed a higher abundance of Macro_FOLR2 + APOE+ cells in the responders compared to the non-responders (Fig. 5e). We also observed the presence of CD4 and CD8 T cells in close proximity to the Macro_FOLR2 + APOE+ cells. Conversely, we did not detect any Macro_FOLR2 + APOE+ cells in the non-responders, indicating a positive association between these cells and treatment response. Furthermore, to confirm the changes in key markers related to myeloid cells before and after neoadjuvant immunotherapy for pan cancer, we collected samples of immunotherapy responses and non-responders for HNSC and CRC. Six markers that showed increased expression levels after treatment in the sequencing results were selected for single channel immunofluorescence staining, including ISG15 (macrophages), NLRP3 (macrophages), IER3 (macrophages), and CD14 (monocytes), CD16 (monocytes), and LAMP3 (DC) (Supplementary Fig. 12). The conclusion related to these three types of myeloid cells can be verified through immunofluorescence chromosomes to confirm the results of multi omics sequencing analysis: Macro_ ISG15, Macro_ NLRP3, Macro_ IER3, Mono_ CD14, Mono_ CD16 and cDC_ The expression level of LAMP3 increases in reactive tumor tissues, and these biomarkers may be one of the response mechanisms of pan cancer immunotherapy, and may also provide a basis for clinical monitoring and efficacy prediction. We will continue to collect more types of samples for verification in the future. These findings align with our overall pan-cancer results and highlight the significance of this macrophage subtype in tumor prognosis.

## Discussion

The pivotal role of TIMs in sculpting the TME, influencing cancer progression, and determining treatment responses has been widely acknowledged. However, our understanding of their variability and adaptability across different cancer types and their impact on immunotherapy remains deficient. To address this gap, we compiled an extensive dataset of tumor specimens that enabled us to comprehensively explore the heterogeneity of TIMs across various cancer scenarios, including treatment-naïve and post-treatment individuals, as well as responders and non-responders to ICB therapy. We integrated expression profiles from diverse cancer types, aiming to uncover shared patterns of cellular variability and treatment responses. Through this thorough analysis, we successfully identified cell types and gene expression patterns that were universally shared, presenting potential broad-spectrum therapeutic targets for multiple cancer types. Importantly, our research sheds light on specific cell types that exhibited significant shifts in proportion upon ICB therapy,

with variations correlating to response status. These findings underscore the remarkable heterogeneity of TIMs and the necessity of considering cancer-specific modifications within myeloid cell populations when designing therapeutic strategies.

Utilizing pseudotime analysis, we observed significant dynamic changes in the kinetic profiles of TIMs, which depended on treatment regimens and exhibited variations between responders and non-responders to ICB therapy. While our findings provide valuable insights, further empirical validation is imperative to bridge the gap between predicted developmental trajectories on a pseudotime scale and the tangible, experimentally observed development of TIMs. Moreover, our investigation revealed substantial disparities in the intra-cellular interactions of TIMs and their interactions with T cells across different response groups. We identified specific receptor-ligand pairs that were intricately linked to response status, shedding light on potential therapeutic targets that are uniquely associated with treatment response. We also discerned both common and distinctive transcription factor signatures within TIMs, response groups, and specific cancer types, these signatures provide a foundation for future investigations to unravel their effects on ICB response.

In addition, we examined the link between the presence of TIMs and patient survival rates, uncovering specific cell types and phenotypic markers that significantly associated with the survival outcomes of patients undergoing ICB therapy. However, it is important to note that our survival analysis was limited to specific cancer types, and broader validation will be necessary to establish the survival impact of the identified TIMs and phenotypic markers across a wider spectrum of cancers. Building upon our findings, we developed a tailored index to forecast the response to ICB therapy. This predictive index could act as a crucial instrument for predicting treatment outcomes based on the phenotypic profiles of TIM subsets in individual patients. Our research serves as a foundation for future studies to expand the utility of this response index to encompass a broader range of cancer types beyond the scope of our current investigation. This endeavor will further enhance our understanding of the intricate relationship between TIMs, phenotypic markers, and treatment response, ultimately guiding personalized therapeutic strategies for improved patient outcomes.

The analysis of the myeloid lineage within the context of cancer immunotherapy response is of utmost importance in identifying specific subsets of myeloid cells that are associated with either favorable or unfavorable responses to immunotherapy. Such insights are invaluable for optimizing treatment strategies. Considering the diverse phenotypes and functions exhibited by myeloid cells, as well as their interactions with other immune cells that profoundly impact treatment outcomes, it becomes crucial to thoroughly investigate the intricacies of the myeloid lineage. This comprehensive exploration allows us to uncover key molecular pathways and signaling mechanisms that govern myeloid cell-mediated immune suppression. By gaining a deeper understanding of these mechanisms, we can develop targeted strategies to overcome immunosuppression and enhance the effectiveness of immunotherapy across various types of cancer. This approach holds great promise in augmenting the efficiency of immunotherapy by harnessing the potential of the myeloid lineage while circumventing its inhibitory effects on the immune system.

## Methods

### Patient recruitment and ethical approval
We collected a total of 192 tumor samples from 129 patients from both published studies and new datasets. These samples consisted of eight cancer types: basal cell carcinoma (BCC), breast cancer, ccRCC, CRC, hepatocellular carcinoma (HCC), head and neck squamous cell carcinoma (HNSCC), intrahepatic cholangiocarcinoma (iCCA), and non-small cell lung cancer (NSCLC)[42–48]. The new datasets enrolled eligible patients meeting the following criteria: (i) CRC stage II (cT3-4, N0) / III

(any cT, N+); (ii) aged ≥18 years old with no gender limitations; (iii) histo-pathologically confirmed CRC of any pathological type; (iv) confirmed by sequencing/immunohistochemistry/polymerase chain reaction to have proficient mismatch repair (pMMR)/microsatellite stable (MSS) status, or deficient mismatch repair (dMMR)/micro-satellite instability-high (MSI-H); (v) those who were immunotherapy-treated received neoadjuvant PD-1 monotherapy; (vi) able to undergo colonoscopy biopsy with a body weight ≥40 kg and a life expectancy ≥6 months. The study excluded individuals who received chemotherapy, targeted therapy, other combined treatment regimens, or Chinese patent medicines during the same cycle. All scRNA data in our study were captured from the 3' end of the transcripts, and VDJ region detection was not performed. This study was conducted in accordance with the Declaration of Helsinki and was approved by the Institutional Review Board of Yunnan Cancer Center (approval number KYCS202199). Informed consent was obtained from all participants whose samples were collected for sequencing, as it was an observational and non-interventional study. The integrity of our study relies on strict adherence to informed consent.

## Treatment and follow-up

The decisions to treat patients with PD-1 inhibitors instead of surgery were made by the primary surgeons of the participating centers. The most common reason for this decision was to avoid multivisceral resection and sphincter dysfunction. As there was no guideline recommending PD-1 inhibitors as neoadjuvant treatment of CRC, the decisions were made based on empirical experience, resulting in non-unanimous regimens. Notably, the study used several kinds of PD-1 inhibitors, including pembrolizumab, sintilimab, camrelizumab, and tislelizumab. Radiographic assessment of response was generally performed after two cycles of treatment, and every 2–3 months thereafter before surgery (if performed), according to the revised RECIST guideline (version 1.1). For patients whose tumor had achieved clinical complete response (cCR; defined as the absence of tumor on radiologic and endoscopic findings), a watch-and-wait approach was offered according to clinical practice.

## Fresh tissue dissociation and single-cell suspension preparation

The colorectal tissues were preserved using GEXSCOPE Tissue Preservation Solution and kept on ice. These samples were washed three times with Hanks' Balanced Salt Solution (HBSS, Gibco, Cat. No.14025-076) and shredded into 1–2 mm pieces. The tissue debris was then subjected to digestion with 2 mL of GEXSCOPE tissue dissociation solution in a 15 mL centrifuge tube (Falcon, Cat. No.352095) with sustained agitation at 37°C for 15 min.

The digested tissues were filtered through a 100 micron filter and washed twice with PBS before surface staining. Anti-CD45 and DAPI were used for immune cell staining at $1 \times 10^6$ cells per mL for 20 min to discriminate live and dead cells after Fc receptor blockade (BioLegend). Viable immune cells (CD45 + DAPI − ) were then sorted and collected on a FACSAria sorter (BD Biosciences).

CD45+ cells were filtered through 40 micron sterile strainers (Falcon, Cat. No.352340) and centrifuged (Eppendorf, 5810 R) at 300 g for 5 min. The supernatant was removed, and the pellets were gently resuspended in 1 mL PBS (HyClone, Cat. No.SA30256.01). To remove red blood cells (RBC), which were frequently a significant portion of the cells produced, 2 mL RBC lysis buffer (Roche, Cat. No. 11 814 389 001) was added to the cell suspension following the manufacturer's protocol. The cells were then centrifuged at $500 \times g$ for 5 min in a microfuge at 15–25 °C and resuspended in PBS (Hyclone, Cat. No.SA30256.01). A trypan blue stain (Bio-Rad, Cat. No.#1450013) was used to count the cells under a microscope (Nikon, ECLIPSE Ts2), and the concentration was adjusted to $1 \times 10^5$ cells/mL. Subsequent sample processing was performed once the cell viability exceeded 80%.

## Single-cell RNA sequencing

**Cell capture and cDNA synthesis.** Using Chromium Single Cell 5 'Library and Gel Bead Kit (10 x Genomics, 1000006) and Chromium Single Cell A Chip Kit (10x Genomics, 120236), the cell suspension (300-600 living cells per uL determined by Count star) was loaded onto the Chromium single cell controller (10 x Genomics) to generate single-cell gel beads in the emulsion according to the manufacturer's protocol. In short, single cells were suspended in PBS containing 0.04% BSA. About 6,000 cells were added to each channel, and the target cells to be recovered were estimated to be about 3,000 cells. Captured cells were lysed and the released RNAs were barcoded through reverse transcription in individual GEMs. Reverse transcription was performed on a S1000TM Touch Thermal Cycler (Bio-Rad) at 53 °C for 45 min, followed by 85 °C for 5 min, and held at 4 °C. The cDNA was generated, amplified, and quality-assessed using an Agilent 4200 (performed by CapitalBio Technology, Beijing).

**Library preparation.** Single-cell RNAseq libraries were constructed using Single Cell 5' Library and Gel Bead Kit, Single Cell V(D)J Enrichment Kit, Human T Cell (10x Genomics, 1000005) and Single Cell V(D)J Enrichment Kit, Human B Cell (10 x Genomics, 1000016) according to the manufacturer's instructions. The libraries were sequenced using an Illumina NovaSeq6000 sequencer with a sequencing depth of at least 100,000 reads per cell with pair-end 150 bp (PE150) reading strategy (performed by CapitalBio Technology, Beijing).

**Quality control and data processing.** De-multiplexing, mapping, and quantification after sequencing were s carried out using Cellranger v6.1.0 (10 x Genomics). Reads were aligned to human genome reference GRCh38 followed by quantification step involving the use of the count function from Cell Ranger, which generated a gene-cell count matrix for each sample. To ensure the quality of the data for both in-house and external datasets, the raw gene-cell count matrix of each sample went through strict quality control assessments. We examined the overall distribution of gene counts in each cell and filtered out cells with gene counts of <200 or >6000. To remove damaged cells indicated by the leakage of mitochondria genes, we evaluated sample-wise percentage of mitochondrial reads sequenced per cell to eliminate potential debris and damaged cells that could affect the overall quality of the data. Cells with over 30% mitochondrial reads were removed from the dataset. Additionally, DoubletFinder[49] was used to identify and remove any doublets present in the data. To maximize the accuracy of doublets identification, we selected an optimal principal component neighborhood size (pK) based on the highest bimodality coefficient (BC) measure for each sample using the mean-variance-normalized bimodality coefficient ($BC_{MVN}$) maximization[49] strategy. Normalization was done using Seurat v4.2.1[50] via natural-log transformation with a scale-factor of 10,000. The resulting normalized data were then scaled and centered. Variation of gene expression across cells was assessed and based on the observed mean and variance of the fitted log-variance to log-mean locally estimated scatterplot smoothing regression model, genes with the highest variance ($n = 2000$) were chosen for downstream analyses. Principal component analysis (PCA) was conducted to obtain the first 50 PCAs which were needed for subsequent analyses.

**Dimension reduction and clustering.** We assessed both the intra- and inter-study variabilities to consider the criteria for integration. We treated each study dataset as a separate batch to ensure that inter-study variabilities were properly addressed during the integration process. Integration was achieved via k-means soft-clustering based on the first 30 PCA coordinates to classify cells into clusters using Seurat SNN combined Harmony v0.1.1[16]. The squared Euclidean distance between PCA embeddings (Z) of cells and their centroids (Y) was calculated to accurately determine the distance between cells and their

respective cluster centroids to determine a correction factor to correct for batch-to-batch variations in each cluster. The centroids were re-calculated and the distance between the cells and their centroids in each cluster were re-measured. This refinement or optimization process was repeated until convergence. Following integration, corrected-PCA embeddings, and harmony embeddings were returned. These harmony embeddings were compared with the original PCA embeddings to evaluate the effectiveness of the integration process (Supplementary Fig. 1a). LISI index was used to evaluate the results of batch correction (Supplementary Fig. 1b).

The first 30 harmony embeddings were used for the UMAP dimension reduction process[50,51]. The Jaccard index was calculated between each cell and $k = 20$ nearest neighbors were chosen to construct a shared neighbor graph (SNN) for the integrated data. To obtain the final clustering assignment, SNN modularity optimization was performed[50]. To identify the DEGs in each cluster in order to conduct cell type annotations, differential expression (DE) analysis was carried out using the Wilcoxon Rank Sum test, to compare each cell cluster with the remaining cells[50]. To correct for multiple-testing, Bonferroni correction was applied. Genes with Bonferroni-adjusted $p < 0.05$ were considered as the DEGs for each cluster.

**Cell type annotation.** Cell type annotation was performed based on the reference of signature markers from literature[6,52], the CellMarker website (http://bio-bigdata.hrbmu.edu.cn/CellMarker/index.html), DEGs obtained for each cell cluster, and the original cell type annotations for the published studies. In-house dataset cell annotations were determined based on the first two criteria and cross-referenced with the cell type annotation from the cells of the published studies present in their clusters. Based on these three sets of reference sources, cell type annotations were finalized and annotated with their signature markers. For further lineage-wise analysis, we classified the myeloid cells into macrophages, monocytes, DCs, and mast cells.

**Stratification of analysis groups.** We analyzed the scRNA-seq data at multiple levels to gain insights into the cellular heterogeneity and response to treatment in different cancer types. The stratifications were conducted at cancer, cell type, and sample type levels. For cancer, analyses were performed at both the pan-cancer and cancer-specific levels. For cell type, analysis was conducted for each cell type independently, or in a lineage-specific way such that, cell types were grouped accordingly to their own lineage types, i.e., macrophage, monocyte, DC, and mast cells. Finally, we also stratified sample types based on their clinical conditions, i.e, into Pre, Post, R, and NR groups for comparison (Fig. 1d). For comparison of sample conditions, the Post (numerator) was compared with the Pre (denominator) and the R (numerator) was compared with the NR (denominator) group. These stratified assessments were conducted for most of the subsequent analyses unless otherwise specified.

**Comparing cell type proportions.** Two-proportions $z$-test was used to compare the proportions of cells between two groups, i.e., the Pre vs. the Post group and the R vs. the NR group. The analysis was performed independently for each cell type and each cancer type. The proportion test was implemented using the prop.test function in R package, which takes two vectors of counts as input, corresponding to the number of successes and failures for each group. In this case, the number of successes corresponds to the number of cells of a particular cell type, while the number of failures corresponds to the total number of cells in the group. The function returns a $p$-value, which indicates the level of statistical significance of the difference in proportions between the two groups. In addition to the proportion test, the fold change of the proportions was also calculated to provide additional information about the magnitude of the difference between the two groups. The fold change was calculated as the ratio

of the proportions in the R or Post group to the proportion in the NR or Pre group, respectively.

**Trajectory analyses.** Within the dynverse[53] framework, we employed a careful selection process to determine the most suitable pseudotime methods for our analysis. After considering various options, we opted to utilize Slingshot[22], Monocle[54], and CytoTRACE[55] for conducting trajectory analyses and robustly assess kinetic changes across different conditions at the pan-cancer level and cancer-specific levels. Cells were ordered along a continuous trajectory based on the similarity of their gene expression profiles and the trajectory was then be used to infer the ordering of cells along a pseudotime. Dimension reduction based on t-SNE was used in Monocle to order cells along a trajectory based on the similarity of their gene expression profiles. Roots were chosen by CytoTRACE and implemented onto Monocle. In this study, the fold change of the average pseudotime orderings for each cell type was calculated for each comparing group (i.e., Post/Pre, and R/NR). We performed $t$-tests to assess the significance of the observed fold changes, $t$-tests were performed. The $p$-value threshold was set to 0.05 significance level. Cells were clustered based on their pseudotime branching states with each cluster sharing a common pseudotime state. DEGs were obtained for each state cluster at BH-FDR < 0.05, and GSEA was conducted based on the DEGs for each sample condition of each lineage type using the hallmark genesets from MSigDB to identify the biological pathways that were differentially regulated across different pseudotime states. Pathways with $p < 0.05$ were retained.

**Cell-cell communication.** Cell-cell interactions were analyzed using the Liana R package[56,57], which allows the identification of ligand-receptor interactions between different cell types, and is a useful resource for understanding the cellular communication networks that underlie the biology of different diseases. We assessed the cell-cell interactions within the myeloid cells and between the myeloid and T cells at both the pan-cancer and cancer-specific levels across sample conditions. T cells were obtained from the same datasets as the myeloid cells, were subjected to the same quality control and filtering steps and were normalized and integrated with the same methods used for the myeloid cells. The cell type annotation for the T-cells was also performed based on their signature genes, DEGs, and annotations from the published studies. Processing and annotating the T-cells in the same way as the myeloid cells enabled direct comparison of the gene expression profiles and cell-cell interactions between these two cell types while minimizing possible bias effects.

Filtering was done based on aggregate rank and interaction frequencies. Aggregate rank is a measure of the overall significance of the interaction between two cell types and was calculated based on the number of ligand-receptor pairs that were present between the two cell types. The lower the aggregate rank, the more significant the interaction. Cell-specific interaction frequency is a measure of the strength of the interaction between two specific cells and was calculated based on the number of ligand-receptor pairs that were present between the two cells. The higher the interaction frequency, the stronger the interaction between the two cells. For filtering, ligand-receptor interactions with an aggregate rank of <0.01 were retained and cell-specific interaction frequencies <75% quantile of the overall frequencies for each sample condition were discarded to ensure the retained interactions were both statistically significant and biologically relevant.

Interaction frequencies for each cell type for different sample conditions were compared within each comparing group (i.e., Post/Pre, R/NR) to obtain specific cell-cell interaction fold-change at pan-cancer and cancer-specific levels. If the absolute fold-change in the interaction frequency was >1, then the cell type was considered to have the greatest change between sample conditions. By comparing the cell-cell interaction frequencies between different sample conditions,

it was possible to identify the cell types and cell-cell interactions that are most affected by changes in the tumor microenvironment or in response to treatment.

**Gene Regulatory Network (GRN) analysis.** The Scenic R package[58] was used to perform GRN analysis. The goal of this analysis was to identify TFs specific to each cell type across different sample conditions at the pan-cancer and cancer-specific levels. GRN analysis involves the identification of the regulatory relationships between TFs and their target genes and the prediction of an activity score for each TF; this activity score measures how actively the TF is involved in the regulation of gene expression in a particular cell and is calculated based on the co-expression patterns of the TF and its target genes. Using Scenic, the TFs that were active in, specific in, or common across cell types were identified by comparing the GRNs across different cell types and sample conditions. To assess the specificity of TF signatures in each cell type, the activity scores of the TFs in each cell type were used to conduct correlation analysis across cancer types and sample conditions. t-test was performed to identify TFs which were significantly differentially expressed between sample conditions but were commonly down- or upregulated in different lineages or across different cancer types based on the TF activity scores.

**Survival analysis to identify survival-specific cell types.** We performed survival analysis to assess the relationship of TIM cell type marker expressions with the survival of cancer patients. Post-anti-PD-L1 treated IMvigor210 UC[36] and post-anti-PD-1 treated ccRCC[37] RNA-seq datasets were used for analyses. The expression levels of cell type marker genes for each sample were calculated using a pre-defined gene set that includes markers for specific cell types. We used this information to categorize each sample into different cell type groups based on the expression of specific cell type markers. Next, we used Kaplan-Meier survival analysis to assess the relationship between cell type marker expression and overall survival of cancer patients. MSRS[59] was used to determine an optimized separation of high and low expression groups. We compared the survival curves of patients with high versus low expression levels of the cell type markers of interest. The significance of the survival differences was assessed using log-rank tests. We also performed Cox proportional hazards regression analysis to assess the impact of cell type marker expression on survival. In addition, we performed sensitivity analysis to assess the robustness of the results to different cutoffs for defining high versus low expression levels of cell type markers.

**Gene-set enrichment analysis (GSEA) and gene ontology (GO) pathway assessment.** For the DEGs of each cell type across strata, genes with Bonferroni-corrected $p < 0.05$ were retained and subjected to GSEA analysis using the biological hallmark, regulatory target, oncogenic signature, and immunologic signature gene sets obtained from MSigDB[23]. GO and KEGG pathway analyses were performed using limma v3.52.4[60]. We further filtered out pathways which were not immune-related and retained those which were unique to each stratum.

**Construction of a treatment response index.** We aimed to construct a treatment response index that reflects the contribution of each cell type to the response to therapy, based on the unique gene sets that are differentially expressed in each cell type group. To achieve this, we identified GSEA pathways that were specifically enriched in each cell type and in each sample response condition. The reference dataset used "c2.all.v2023.1.Hs.symbols.gmt", "c6.all.v2023.1.Hs.symbols.gmt" and "h.all.v2023.1.Hs.symbols.gmt" from the MSigDB database. We used the enrichment scores for each pathway as weight coefficients for their cell type proportions in each sample of each cancer type. We then multiplied the proportion of each cell type by the enrichment score of

each pathway in the corresponding cell type group and summed the results across all cell types to obtain a treatment response index for each sample. The final response index for each sample is the sum of all weighted cell type proportions in the sample. This index reflects the overall contribution of each cell type to the response to therapy, based on the unique gene sets that are enriched in each cell type group.

## Reporting summary
Further information on research design is available in the Nature Portfolio Reporting Summary linked to this article.

## Data availability
The processed single-cell RNA-sequencing data have been deposited in the National Genomics Data Center (NGDC) under the accession number PRJCA016919. All other data are available in the main text or in the supplementary materials or from the corresponding author upon request. Source data are provided with this paper.

## Code availability
Source code can be found in the GitHub repository under: https://github.com/pancancer/myeloid.

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

## Acknowledgements

The computations and data handling were enabled by resources at the Uppsala Multidisciplinary Center for Advanced Computational Science (UPPMAX). We thank Dr. Daniel Ackerman of Insight Editing London for editing this manuscript during preparation. We would also like to acknowledge The SCA Consortium for their valuable contributions and

support to this project. This work is supported by the following fundings: (1)National Natural Science Foundation of China (National Science Foundation of China)—81860499 [Chunjie Xiao]; (2)Karolinska Institute Network Medicine Global Alliance Collaborative Grant C24401073 [Xuexin Li & Lu Pan]; (3)Yunnan revitalization talent support program −2023-KHRCBZ-B08 [Weiyuan Li].

## Author contributions

Conceptualization: X.L., L.P., W.L. Methodology: X.L., L.P., W.L., and X.Z. Data analysis: L.P. Supervision: X.L., C.X. Manuscript writing: L.P., X.L., W.L, F.G. Reviewing and editing: all authors.

## Funding

## Competing interests

The authors declare no competing interests.
