## [Peer Review File · Nature Communications]

A Single-Cell Pan-cancer analysis to show the variability of Tumor-Infiltrating Myeloid Cells in Immune Checkpoint BlockadeEditorial Note: This manuscript has been previously reviewed at another journal that is not operating a transparent peer review scheme. This document only contains reviewer comments and rebuttal letters for versions considered at *Nature Communications*.

REVIEWERS' COMMENTS

Reviewer #3 (Remarks to the Author):

The authors have addressed all points related to the previous review, the manuscript has significantly improved and should be accepted as is.

Reviewer #4 (Remarks to the Author):

Based on my review of the answers to these questions, I have my major summary as the following:

1. For major question 1, R2 raised two sub-questions: one is why the authors focused on myeloid cells (how the authors selected for myeloid cells), the other one is the number of cells they analyzed. The authors didn't answer "how the authors selected for myeloid cells" in their response.
2. R2 request a waffle plot to show TIM proportion change in different conditions. The authors didn't show waffle plot, instead they provided a piechart (figure 1d) as well as fraction barplot (figure 1e) to show this change. The current piechart is informative to me.
3. I have no further question regarding this and think the authors have answered this question comprehensively. One minor comment is: it's better to add CD14/CD68 and CD163 to the bubble plot for the macrophages populations.
4. Thanks for the additional work done by the authors to confirm. Applying mitochondrial percentage filtration is a very common operation in scRNAseq analysis. While I agree and recommend apply mitochondrial percentage filtration as did by the authors. Though, the example provided by the authors is not proper cause Zhang's group focused on NK population, while here the authors is focused on myeloid cells, and there may exist myeloid population that has low transcriptome complexity (and thus relatively higher mito fraction, like neutrophils).

5. I agree with R2 that batch effect evaluation is an important topic for most scRNAseq studies. But I don't agree with R2 comment stating that Harmony, in general, is an unreliable method. Regarding the authors answer to this question, please update your Figure S1 legends to clearly describe what's showed in current Figure S1b. it seems that the current Figure S1b legend is not updated.

6. I think the reviewer's comment was answered and I have no further comment.

7. I think the authors didn't answer the reviewer's comment. The reviewer would like the authors to perform a comparison between their FOLR2 mac versus the macrophage subset reported from this paper : <https://doi.org/10.1126/sciimmunol.abf7777>.

Minor points:

1. The authors didn't answer the reviewer's comment about "taking into account their tumor types".

2. The current displayed content is clear to me and I have no further comment.

Reviewer #5 (Remarks to the Author):

The authors addressed most of the issues raised in the original review satisfactorily and thereby considerably improved the manuscript. Specifically, the authors carried out additional computational analyses (e.g. functional enrichment analysis, LISI index, pseudotime analysis using several tools as well as a number of minor issues) and experimental validation using patient samples.

Response to all reviewers

Reviewer #3:

The authors have addressed all points related to the previous review, the manuscript has significantly improved and should be accepted as is.

Response: Thank you for your reviews and comments. We are pleased to have addressed the issues you raised. Once again, thank you!

Reviewer #4:

Based on my review of the answers to these questions, I have my major summary as the following:

1. For major question 1, R2 raised two sub-questions: one is why the authors focused on myeloid cells (how the authors selected for myeloid cells), the other one is the number of cells they analyzed. The authors didn't answer "how the authors selected for myeloid cells" in their response.

Response: Thank you for your reviews. We apologize for not addressing "how the authors selected for myeloid cells" in our previous response. In fact, we have explained "why we chose myeloid cells" in the main section. Additionally, we have detailed the process of cell annotation and "how we selected myeloid cells" in the methods section. The specific details are as follows:

Main: The development of immune-checkpoint blockade (ICB) immunotherapy marked a significant breakthrough in the field of cancer treatment, yielding promising clinical outcomes in recent years. While T lymphocytes have been the primary focus of ICB research, myeloid cells, have not received comparable attention despite their abundance and versatility within the tumor microenvironment (TME). There is growing interest in investigating tumor-infiltrating myeloid cells (TIMs), but their influence within the TME and their impact on ICB responses remain incompletely understood.

Methods: Cell type annotation. Cell type annotation was performed based on the reference of signature markers from literature^{54,55}, the CellMarker website (<http://bio-bigdata.hrbmu.edu.cn/CellMarker/index.html>), DEGs obtained for each cell cluster, and the original cell type annotations for the published studies. In-house dataset cell annotations were determined based on the first two criteria and cross-referenced with the cell type annotation from the cells of the published studies present in their clusters.

Based on these three sets of reference sources, cell type annotations were finalized and annotated with their signature markers. For further lineage-wise analysis, we classified the myeloid cells into macrophages, monocytes, DCs, and mast cells.

2. R2 request a waffle plot to show TIM proportion change in different conditions. The authors didn't show waffle plot, instead they provided a piechart (figure 1d) as well as fraction barplot (figure 1e) to show this change. The current piechart is informative to me.

Response: Thank you very much for your reviews. We are pleased that you accepted the information presented in the pie chart. Thank you again!

3. I have no further question regarding this and think the authors have answered this question comprehensively. One minor comment is: it's better to add CD14/CD68 and CD163 to the bubble plot for the macrophages populations.

Response: Thank you very much for your suggestion. Regarding your suggestion, initially, we did include CD14/CD68 and CD163 in the bubble plot for the macrophage group. However, considering our new macrophage classification method needed to be distinguished from the traditional M1-like/M2-like macrophage classification (where CD68/CD163 expression is used to differentiate M1-like/M2-like macrophages), we ultimately chose not to display the bubble plot for CD68/CD163. Nevertheless, this does not affect our analysis results, as CD68/CD163 were important reference markers during the initial screening of macrophages. Thank you again for your suggestion, which has made our research more rigorous.

4. Thanks for the additional work done by the authors to confirm. Applying mitochondrial percentage filtration is a very common operation in scRNAseq analysis. While I agree and recommend apply mitochondrial percentage filtration as did by the authors. Though, the example provided by the authors is not proper cause Zhang's group focused on NK population, while here the authors is focused on myeloid cells, and there may exist myeloid population that has low transcriptome complexity (and thus relatively higher mito fraction, like neutrophils).

Response: Thank you for your suggestion. As you mentioned, filtering by mitochondrial percentage may lead to the exclusion of myeloid cell subgroups with low transcriptomic complexity, such as neutrophils. Since the datasets included in this study were all subjected to scRNA-seq using 10x Genomics technology, we consulted with 10x official technical personnel to find alternative filtering methods that consider the presence of neutrophils. We learned that the updated Chromium GEM-X Single Cell Gene Expression v4 (updated in 2024) addresses the issue of capturing neutrophils from a technical standpoint. However, the 10x Genomics Chromium GEM-X technology

used in this study rarely captures neutrophils. After further consultation with 10x technical personnel, we confirmed the reliability of using mitochondrial percentage for cell filtering in our study. Additionally, another study on pan-cancer myeloid cells by Zhang Zemin et al. employed the same method (doi: 10.1016/j.cell.2021.01.010). Therefore, we believe that the methodology used in this study is reasonable. Thank you again for your suggestion, and we hope our response satisfies your concerns.

5. I agree with R2 that batch effect evaluation is an important topic for most scRNAseq studies. But I don't agree with R2 comment stating that Harmony, in general, is an unreliable method. Regarding the authors answer to this question, please update your Figure S1 legends to clearly describe what's showed in current Figure S1b. it seems that the current Figure S1b legend is not updated.

Response: Thank you very much for your suggestion. We have updated the legend of Figure S1b as follows:

(b) The line graph illustrates the density performance of the LISI index across different batch algorithms.

Thank you again for your suggestion, which has made our research more rigorous.

6. I think the reviewer's comment was answered and I have no further comment.

Response: Thank you very much for your comments, we have learned a lot from this revision process.

7. I think the authors didn't answer the reviewer's comment. The reviewer would like the authors to perform a comparison between their FOLR2 mac versus the macrophage subset reported from this paper : <https://doi.org/10.1126/sciimmunol.abf7777>.

Response: Thank you for your suggestion. We highly value your advice and have compared our FOLR2 macrophages with the macrophage subpopulations identified in Sarah et al.'s study. However, Sarah et al.'s study conducted scRNA-seq analysis on mouse tissues (as indicated by the red box in the figure below), preventing us from integrating or reference mapping their scRNA-seq data with ours. Therefore, we can only compare FOLR2 macrophages with other macrophage subtypes within our own dataset to elucidate the potential roles and significance of FOLR2 macrophages. Compared to Sarah et al.'s study, the pan-cancer macrophage subgroups defined in our research hold broader significance. Comparing FOLR2 macrophages with other macrophage subtypes in our data enables a more comprehensive understanding of FOLR2 macrophages. We hope this explanation meets your satisfaction. Once again, we sincerely thank you for your assistance; your suggestions have greatly enhanced our confidence in this study.

Status	Public on Jan 07, 2022
Title	Three tissue resident macrophage subsets co-exist across organs with conserved origins and lifecycles
Organisms	Homo sapiens ; Mus musculus
Experiment type	Expression profiling by high throughput sequencing
Summary	Resident macrophages orchestrate homeostatic, inflammatory, and reparative activities. It is appreciated that different tissues instruct specialized macrophage functions. However, individual tissues contain heterogeneous subpopulations, and how these subpopulations are related is unclear. We asked whether common transcriptional and functional elements could reveal an underlying framework across tissues. Using single-cell RNA sequencing and random forest modeling, we observed that four genes could predict three macrophage subsets that were present in murine heart, liver, lung, kidney, and brain. Parabiotic and genetic fate mapping studies revealed that these core markers predicted three unique life cycles across 17 tissues. TLF+ (expressing TIMD4 and/or LYVE1 and/or FOLR2) macrophages were maintained through self-renewal with limited monocyte input; CCR2+ (TIMD4–LYVE1–FOLR2–) macrophages were almost entirely replaced by monocytes, and MHC-IIhi macrophages (TIMD4–LYVE1–FOLR2–CCR2–), while receiving modest monocyte contribution, were not continually replaced. Rather, monocyte-derived macrophages contributed to the resident macrophage population until they reached a defined upper limit after which they did not outcompete pre-existing resident populations. TLF+ macrophages were first to emerge in the yolk sac and early fetal organs. Fate mapping studies in the mouse and human single-cell RNA sequencing indicated that TLF+ macrophages originated from both yolk sac and fetal monocyte precursors. Furthermore, TLF+ macrophages were the most transcriptionally conserved subset across mouse tissues and between mice and humans, despite organ- and species-specific transcriptional differences. Here, we define the existence of three murine macrophage subpopulations based on common life cycle properties and core gene signatures, and suggest a common starting point to understand tissue macrophage heterogeneity.
Overall design	Single cell RNA sequencing of tissue macrophages in 5 tissues in mice.
Contributor(s)	Hamidzada H , Epelman S
Citation(s)	Dick SA, Wong A, Hamidzada H, Nejat S et al. Three tissue resident macrophage subsets coexist across organs with conserved origins and life cycles. Sci Immunol 2022 Jan 7;7(67):eabf7777. PMID: 34995099
Submission date	Nov 11, 2021
Last update date	Apr 08, 2022
Contact name	Slava Epelman
E-mail(s)	slava.epelman@uhn.ca
Organization name	University Health Network
Street address	101 College St.
City	Toronto
State/province	Ontario
ZIP/Postal code	M5G 1L7
Country	Canada
Platforms (2)	GPL16791 Illumina HiSeq 2500 (Homo sapiens) GPL17021 Illumina HiSeq 2500 (Mus musculus)
Samples (14)	GSM5687566 Sample 1_Mouse heart macrophages GSM5687567 Sample 2_Mouse liver macrophages GSM5687568 Sample 3_Mouse lung macrophages GSM5687569 Sample 4_Mouse kidney macrophages GSM5687570 Sample 5_Mouse brain macrophages GSM5687571 Sample 6_Parabiosis heart macrophages_host_5wks

Minor points:

1. The authors didn't answer the reviewer's comment about "taking into account their

tumor types”.

Response: Thank you for your suggestion. As you indicated, in Figure S1c, we have already presented the number of cells included for different tumor types. Additionally, in Figure 1e, we have shown the proportions of various cell types across different tumor types. These considerations account for the diversity of tumor types. Once again, we appreciate your suggestion!

2. The current displayed content is clear to me and I have no further comment.

Response: Thank you for your reviews and comments. We are pleased to have addressed the issues you raised. Once again, thank you!

Reviewer #5:

The authors addressed most of the issues raised in the original review satisfactorily and thereby considerably improved the manuscript. Specifically, the authors carried out additional computational analyses (e.g. functional enrichment analysis, LISI index, pseudotime analysis using several tools as well as a number of minor issues) and experimental validation using patient samples.

Response: Thank you for your reviews and comments. We are pleased to have addressed the issues you raised. Once again, thank you!

Other Revise:

We have added a co-first author, Weifeng Hong, who oversaw the entire revision process (including the first and second revisions). This decision was unanimously agreed upon by all of our authors.